# PET Radiomics and Response to Immunotherapy in Lung Cancer: A Systematic Review of the Literature

**DOI:** 10.3390/cancers15123258

**Published:** 2023-06-20

**Authors:** Laura Evangelista, Francesco Fiz, Riccardo Laudicella, Francesco Bianconi, Angelo Castello, Priscilla Guglielmo, Virginia Liberini, Luigi Manco, Viviana Frantellizzi, Alessia Giordano, Luca Urso, Stefano Panareo, Barbara Palumbo, Luca Filippi

**Affiliations:** 1Department of Biomedical Sciences, Humanitas University, Via Rita Levi Montalcini 4, 20072 Pieve Emanuele, Italy; 2IRCCS Humanitas Research Hospital, Via Manzoni 56, 20089 Rozzano, Italy; 3Nuclear Medicine Department, E.O. “Ospedali Galliera”, 16128 Genoa, Italy; francesco.fiz@galliera.it; 4Nuclear Medicine Department and Clinical Molecular Imaging, University Hospital, 72076 Tübingen, Germany; 5Unit of Nuclear Medicine, Biomedical Department of Internal and Specialist Medicine, University of Palermo, 90100 Palermo, Italy; riclaudi@hotmail.it; 6Department of Engineering, Università degli Studi di Perugia, Via Goffredo Duranti, 06125 Perugia, Italy; francesco.bianconi@unipg.it; 7Nuclear Medicine Unit, Fondazione IRCCS Ca’ Granda, Ospedale Maggiore Policlinico, 20122 Milan, Italy; angelo.castello@policlinico.mi.it; 8Nuclear Medicine Unit, Veneto Institute of Oncology IOV—IRCCS, 35128 Padua, Italy; priscilla.guglielmo@iov.veneto.it; 9Nuclear Medicine Department, S. Croce e Carle Hospital, 12100 Cuneo, Italy; v.liberini@gmail.com; 10Medical Physics Unit, Azienda USL of Ferrara, 45100 Ferrara, Italy; luigi.manco@ausl.fe.it; 11Department of Radiological Sciences, Oncology and Anatomo-Pathology, Sapienza University of Rome, 00161 Rome, Italy; viviana.frantellizzi@uniroma1.it; 12Nuclear Medicine Unit, IRCCS CROB, Referral Cancer Center of Basilicata, 85028 Rionero in Vulture, Italy; alessia.giordano@crob.it; 13Department of Nuclear Medicine PET/CT Centre, S. Maria della Misericordia Hospital, 45100 Rovigo, Italy; luca.urso@unife.it; 14Nuclear Medicine Unit, Oncology and Haematology Department, University Hospital of Modena, 41124 Modena, Italy; panareo.stefano@aou.mo.it; 15Section of Nuclear Medicine and Health Physics, Department of Medicine and Surgery, Università degli Studi di Perugia, 06125 Perugia, Italy; barbara.palumbo@unipg.it; 16Nuclear Medicine Section, Santa Maria Goretti Hospital, 04100 Latina, Italy; lucfil@hotmail.com

**Keywords:** immunotherapy, lung cancer, PET, response to therapy, PD-L1

## Abstract

**Simple Summary:**

The present review was performed in order to provide a comprehensive overview of the existing literature concerning the applications of positron emission tomography (PET) radiomics in lung cancer patients candidates or those currently undergoing immunotherapy. Fifteen papers were included, thirteen were qualified as using conventional radiomics approaches, and two used deep learning radiomics. Different settings were analyzed, from the utility of radiomics as an additional tool for predicting the expression of PD-L1 or the tumor microenvironment, to the utility of artificial intelligence in evaluating the response to immunotherapy. Although radiomics seems promising in these fields, too limited data are now available. Indeed, the first limitation is the low amount of data, heterogeneity in the provided information, the still limited experience and also the small amount of expertise in this field. Therefore, radiomics is still far from to be considered for daily routine clinical practice, although some additional efforts are required for the next future, mainly in patients scheduled or undergoing immunotherapy.

**Abstract:**

The aim of this review is to provide a comprehensive overview of the existing literature concerning the applications of positron emission tomography (PET) radiomics in lung cancer patient candidates or those undergoing immunotherapy. Materials and Methods: A systematic review was conducted on databases and web sources. English-language original articles were considered. The title and abstract were independently reviewed to evaluate study inclusion. Duplicate, out-of-topic, and review papers, or editorials, articles, and letters to editors were excluded. For each study, the radiomics analysis was assessed based on the radiomics quality score (RQS 2.0). The review was registered on the PROSPERO database with the number CRD42023402302. Results: Fifteen papers were included, thirteen were qualified as using conventional radiomics approaches, and two used deep learning radiomics. The content of each study was different; indeed, seven papers investigated the potential ability of radiomics to predict PD-L1 expression and tumor microenvironment before starting immunotherapy. Moreover, two evaluated the prediction of response, and four investigated the utility of radiomics to predict the response to immunotherapy. Finally, two papers investigated the prediction of adverse events due to immunotherapy. Conclusions: Radiomics is promising for the evaluation of TME and for the prediction of response to immunotherapy, but some limitations should be overcome.

## 1. Introduction

Lung cancer (LC) is the leading cause of cancer-related death worldwide and represents a serious threat for public health [1] despite advances in diagnosis and therapy [2,3]. Surgical resection is the standard of care for LC patients at stages I and II, and for many years, platinum-based chemotherapy has represented a mainstay for the management of patients with extensive disease [4]. Recently, the therapeutic landscape has been thoroughly changed by the implementation of immune-checkpoint inhibitors (ICIs).

In metastatic non-small cell lung cancer (NSCLC) harboring driver mutations (e.g., EGFR, ALK, or ROS1), targeted therapies are usually preferred over other approaches since they have been found effective and to have tolerable toxicity. Nevertheless, even in the absence of driver mutations, ICIs should be considered, alone or in combination with chemotherapy, as a valuable option [4]. In 2015, a phase III comparative study showed that nivolumab, a monoclonal antibody directed towards PD-1, provided a significant benefit in terms of prolonged overall survival (OS), with respect to docetaxel, in squamous and non-squamous NSCLC submitted to immunotherapy as a second-line regimen [5]. These initially encouraging results were further confirmed in subsequent clinical trials, leading to the implementation of ICIs for the management of patients with advanced NSCLC [6]. However, it has to be underlined that only 50% of LC patients will show a response to immunotherapy [7]. Furthermore, immune-related adverse events (irAEs), namely the side effects of ICIs treatment potentially occurring in any organ or system with a wide spectrum of severity, affect up to 76% of the patients under immunotherapy and can represent an important cause of treatment discontinuation [8]. From this perspective, there is an unmet need for laboratory and imaging biomarkers suitable for identifying LC patients who are more likely to benefit from ICIs.

Positron emission tomography/computed tomography (PET/CT) with fluorine-18-fluorodeoxyglucose ([18F]FDG) has a well-established role in staging and response assessment in many oncological conditions [9]. As concerns patients’ prognostic stratification before immunotherapy, [18F]FDG PET/CT has been applied with interesting preliminary results since some PET-derived parameters, such as whole-body metabolic tumor volume (wbMTV) and total lesion glycolysis (wbTLG), were found to be predictors of response [10,11].

In recent years, radiomics, an emerging discipline based on the quantitative analysis of imaging data, has been gaining ever-increasing attention for its capability through machine learning analysis to generate predictive models [12]. The main scope of radiomics is to extract from medical images the quantitative data (i.e., features) that are undetectable to the human eye and are able to be reproduced, interpreted, and correlated with some clinical endpoints (e.g., response to therapy, treatment failure, survival, etc.). In this regard, pre-clinical and clinical studies suggest the potential of PET radiomics for the prediction of immunotherapy response [13,14,15].

The aim of the present systematic review is to provide a comprehensive overview of the existing literature concerning the applications of PET radiomics in LC patients candidates or undergoing ICIs and to outline the most relevant issues emerging from data analysis, delineating potential next steps for its widespread implementation in clinical practice.

## 2. Materials and Methods

### 2.1. Research Strategy and Study Selection

A systematic review was conducted in accordance with the preferred reporting items for systematic reviews guidelines (PRISMA) by R.L., L.U., and P.G. [16]. The authors ran queries to retrieve prospective or retrospective studies on the use of radiomics application on PET images of immunotherapy as applied to lung cases on databases and web sources (i.e., *PubMed*, *Google Scholar*, and *Scopus*). The search was carried out on 12th March 2023, using the multiple queries reported here: “(lung cancer OR NSCLC) AND (immune checkpoint inhibitors OR immunotherapy OR ICI) AND (radiomics OR features) AND pet NOT review”, “pet AND (radiomic OR radiomics OR texture) AND lung AND immunotherapy”, “pet AND (radiomics OR texture) AND lung AND ICI”, “pet AND (artificial intelligence OR deep learning) AND lung AND immunotherapy”, “pet AND (radiomic OR deep learning) AND lung”. English-language original articles were considered. The review was registered on the PROSPERO database with the number CRD42023402302.

The title and abstract were independently reviewed by three authors (P.G., L.E., and L.F.) to evaluate study inclusion. Full articles were retrieved when the abstract was considered relevant. Duplicate, out-of-topic, and review papers or editorial articles and letters to editors were excluded.

### 2.2. Radiomics Methodology and Study Quality

Radiomics approaches were divided into two groups: the conventional ones (also referred to as “hand-crafted”) and those based on deep learning [17]. Conventional radiomics involved the delineation of the region of interest (ROI) (which can be manual, semi-automated, or fully automated) and the subsequent extraction of a set of pre-defined parameters such as first-order statistics, shape, and texture features [18]. The features are eventually fed to some classification and/or regression model to produce predictions about the clinical endpoint investigated. By contrast, deep learning radiomics makes use of computational architectures (convolutional neural networks—CNNs) in which the features are no longer defined a priori but learned from the data. In this scheme, the detailed lesion delineation is not strictly necessary, as this step is replaced by approximate localization, which is generally achieved by defining a fixed-shape (typically square) bounding box around the suspicious area. Furthermore, CNNs have internal classification blocks (fully connected layers), which makes external classifiers unnecessary.

For each study, the radiomics analysis was assessed based on the radiomics quality score (RQS 2.0 https://www.radiomics.world/rqs2 (accessed on 1 April 2023)) introduced by Lambin and colleagues in 2017 [19] to specifically evaluate the quality of reporting in the radiomics context. RQS 2.0 consists of 36 checkpoints that reward or penalize radiomics studies to encourage best scientific practices. For a robust calculation, RQS 2.0 was assessed by a multidisciplinary panel of three raters, respectively, namely one nuclear medicine physician (F.F.), one medical physicist (L.M.), and one engineer (F.B.), all with at least 5 years of experience in radiomics. After a preliminary training session to calibrate the methodology and the scoring system, each rater read, assessed, and scored the papers independently. Once this step was completed, the evaluation panel reconvened for a final joint session in which a consensus score was assigned to each paper.

## 3. Results

### 3.1. Radiomics Assessment

In total, 15 papers were selected (Table 1).

In Figure 1, we report the PRISMA statement, while the RQS v.2 is reported in Table 2.

Of the 15 included papers, 13 were qualified as using conventional radiomics approaches and 2 as using deep learning radiomics [24,28].

Within the conventional radiomics group, semi-automated segmentation was the most common approach to ROI identification (10 studies), followed by manual delineation (three). Multiple-ROI identification with the assessment of feature robustness to inter-observer variability was carried out in 4 studies out of 15. The total number of features initially extracted from each ROI ranged from 6 [25] to 3488 [20]; the most common features were PET semi-quantitative parameters, first-order statistics, and texture features. Feature selection was performed in the majority of the studies either through one single method or a combination of them. The least absolute shrinkage and selection operator (LASSO) was the most popular approach for this task (nine studies), followed by redundancy analysis via Pearson’s correlation coefficient (five). Multivariate logistic regression was the preferred model for endpoint prediction (six studies); other approaches were weighted linear models (three) and random forest classification (two).

The two papers investigating deep learning were based, respectively, on a two-dimensional small residual convolutional network (SresCNN [27]) and a three-dimensional CNN [25]. Specifically, in [36], the input to the network was a series of planar boxes from consecutive slices clipped around the ROI, whereas in [24], it was a cube-shaped volume around the inspected lesion. Data augmentation was used in both papers.

### 3.2. Baseline PET for the Prediction of Biomarker Expression

PD-L1 expression status and tumor mutational burden are considered both valuable predictive factors in NSCLC-patient candidates of ICIs. Nevertheless, a variable number of patients cannot benefit from ICIs, independently of PD-L1 score, via an unknown mechanism [36], thus requiring the identification of additional predictive parameters. In this context, the rising role of radiomics could contribute to identifying novel biomarkers useful for the correct identification of patients who will most likely benefit from ICIs.

Several papers have investigated the potential role of radiomics to predict PD-L1 expression status in NSCLC patients before starting ICI treatment [20,26,28,35]. Jian et al. [20] extracted radiomics features from the [18F]FDG PET/CT of 399 NSCLC patients. After reduction with the LASSO algorithm, three predictive models were developed based on features from CT alone, PET alone, and PET/CT-combined images. For PD-L1 evaluated by using the SP142 kit, the AUCs for predicting PD-L1 > 1% vs. PD-L1 > 50% were 0.97 vs. 0.80, 0.61 vs. 0.65, and 0.97 vs. 0.77, respectively, for CT alone, PET alone, and PET/CT-combined images. On the other hand, for PD-L1 evaluated with 28-8 kit, AUC was 0.86, 0.62, and 0.85 for predicting PD-L1 > 1% and 0.91, 0.75, and 0.88 for predicting PD-L1 > 50%, respectively, for CT alone, PET alone, and PET/CT-combined images.

Li et al. [26] proposed a combined model between radiomics features extracted from [18F]-FDG PET/CT and clinicopathologic variables (i.e., age, gender, tumor location, histology type and grade, carcinoembryonic antigen level, smoking history, and Ki-67). Overall, 255 patients were enrolled and divided into training (*n* = 170) and validation groups (*n* = 85). Eighteen out of eighty radiomics features (six from CT and twelve from PET) were useful for predicting PD-L1 > 1%, and seven (four from CT and three from PET) for PD-L1 > 50%. The combined model for the prediction of PD-L1 > 1% showed an AUC score of 0.757 (95% CI: 0.699–0.808), whereas for the prediction of PD-L1 > 50%, the AUC was 0.814 (95% CI: 0.761–0.860).

Similarly, Zhao et al. [35] built and validated a radiomics model, a clinical model, and their combination for predicting PD-L1 expression status (if ≥ 1%) in NSCLC patients. After the LASSO algorithm and 10-fold cross-validation, two optimal radiomics features (Gray-level run-length matrix (GLRLM)_Run percentage (RP) and Shape_Sphericity) were selected. The AUC values of the combined model were significantly higher than those of the clinical model both in the training (0.718 vs. 0.638, *p* = 0.004) and validation group (0.769 vs. 0.640, *p* = 0.007), while there were no significant differences between the combined and radiomics models in both the training and validation cohorts. Hence, based on the combined model, an individualized nomogram was developed, showing good consistency between the predictive probability and the actual predicted probability in the training group (χ^2^ = 1.463, *p* = 0.481) and the validation group (χ^2^ = 1.563 *p* = 0.458), with no significant differences between different PET/CT scanners.

Moreover, in a recent study, Mu et al. [28] investigated the potential role of a deeply learned score (DLS) for predicting PD-L1 expression status in 697 NSCLC patients. The results were satisfying, with an AUC ≥ 0.82 for discriminating PD-L1-positive (if ≥ 1%) vs. -negative patients. Interestingly, DLS-paired immunohistochemistry derived the PD-L1 status for predicting progression-free survival (PFS) and OS.

The characterization of the tumor microenvironment (TME), including tumor-infiltrating lymphocytes (TILs) CD3+ and CD8+, represents another independent biomarker, even though the heterogeneity between primary tumor and metastatic lesions as well as the difficulty of biopsy in some patients makes its use unsatisfactory for monitoring the efficacy of ICIs [34,37]. Two studies explored the potential role of radiomics to predict the composition of TME [24,31]. Park et al. [24] developed a deep learning model to estimate the TME in lung adenocarcinoma using data from the [18F]FDG PET/CT and RNA sequencing of 93 patients. The cytolytic activity score (CytAct) was used as an indirect biomarker for TME, as it represents CD8+ T-cell activity, and it is easy to calculate. The model was validated in two independent cohorts (*n* = 43 and *n* = 16, respectively) and showed a positive correlation with the CytAct of RNA sequencing from both (rho = 0.32 and 0.47, respectively). On the other hand, in the ICI cohort, the predicted CytAct was inversely correlated with tumor size after ICI treatment (rho = −0.54). In addition, a higher minimum predicted CytAct was also associated with prolonged PFS and OS (HR 0.25, *p* = 0.001 and HR 0.18, *p* = 0.004, respectively).

More recently, Tong et al. [31] applied a machine learning model to evaluate the TME phenotype combining [18F]FDG PET/CT data and clinical characteristics of NSCLC patients from Daping Hospital (DPH) and The Cancer Imaging Archive (TCIA). First, the Delong test demonstrated that the PET/CT model outperformed the CT alone model to predict the CD8 expression. Later, PET/CT radiomics clinical model, integrating significant clinical features with Rad score, was able to predict TME status in NSCLC (training AUC of 0.932 and testing AUC of 0.920), showing better performance compared to the clinical and radiomics models separately (AUC = 0.932 vs. 0.868 vs. 0.907, respectively). In addition, the radiomics clinical combined model was also applied in the TCIA cohort for predicting the TME phenotype. Based on the combined model, patients were classified into two predicted CD8 groups (high vs. low). The first group showed a significantly higher immune score and more activated immune pathways than the second group, implicating better response when treated with ICI.

### 3.3. The Prediction of Response to Immunotherapy

Radiomics could contribute both to predicting the response to immunotherapy and identifying therapy-responsive patients accurately and early: from the clinical point of view, each additional month of ineffective therapy can be crucial for metastatic NSCLC patients and costly for public health.

Thus far, the radiomics potentialities from baseline PET as predictive parameters have been tested by two authors [21,28]. Mu et al. [28] published one of the first studies investigating the predictive role of radiomics in NSCLC patients treated with ICI. The authors extracted radiomics features from baseline CT, PET, and PET/CT-fused images and found that characteristics related to heterogeneity (i.e., short-run low gray emphasis or short-zone emphasis) could reliably predict durable benefit from ICI treatment (AUC of 0.86 for training, 0.83 for retrospective, and 0.81 for prospective test cohorts). Nevertheless, the main limitation of this study was the lack of PD-L1 expression data for many enrolled patients, hindering a direct comparison of the authors’ model with the PD-L1 status. Polverari et al. [21] found that NSCLC patients with elevated TLG, volume, and high tumor heterogeneity in asymmetry (i.e., skewness) and kurtosis were more likely to experience disease progression during ICI treatment, although the lack of a robust validation cohort represents the main limitation of this study. 

Radiomics features from serial PET/CT scans before, during, or after ICIs were assessed by three authors [25,30,32]. In 2020, Valentinuzzi et al. [25] created a [18F]FDG PET radiomics signature (iRADIOMICS) consisting of the most promising radiomics features extracted by the [18F]-FDG images and able to predict the response of metastatic NSCLC (stage IV) to pembrolizumab compared to the clinical standards (PD-L1 immunohistochemistry and iRECIST). Thirty patients receiving pembrolizumab were scanned with [18F]FDG PET/CT at baseline and months 1 and 4. Response to therapy was defined as OS > 14.9 months. iRADIOMICS (baseline), iRECIST (months 1 and 4), and PD-L1 (baseline) signatures were constructed using univariate or multivariate logistic regression analyses. At baseline PET, none of the standard volume-based features (volume and SUVmax) were able to discriminate responders from non-responders. On the contrary, the predictive power of the baseline iRADIOMICS signature was higher than the PD-L1 signature (AUC of 0.81 vs. 0.60) and comparable to month 1 and month 4 iRECIST signatures (AUC of 0.79 and 0.81, respectively), allowing earlier identification of the response by at least one month. To further validate the predictive ability of all models, the accuracy of predictions was calculated using 5-fold cross-validation. Multivariate baseline iRADIOMICS was found to be superior to the current standards (PD-L1 and iRECIST signatures). Both conventional parameters and radiomics features extracted from the month 1 and month 4 PET/CT images were not significantly different between responders and non-responders, with the only exception of the volume of the lesion after 1 month from the start of ICIs (*p* = 0.035, AUC = 0.75 (0.55–0.95)). Despite these promising data, authors analyzed only primary tumors yet neglected lymph nodes (LN) and distant metastases (DM).

Tankyevych et al. [30] retrospectively evaluated 83 patients with locally advanced or metastatic NSCLC treated with immunotherapy. They aimed to assess the ability of radiomics features from baseline (PET/CT0) and both early (PET/CT1 = 6–8 weeks after the initiation of therapy) and late (PET/CT2 = 3 months after the initiation of therapy) follow-up [18F]FDG PET/CT scans as well as their evolution (delta radiomics) to predict durable clinical benefit (DCB), progression (according to PERCIST at the first restaging and iPERCIST and RECIST1.1 after 3 months of treatment), response to therapy, PFS, and OS. Seven multivariate models with different combinations of clinical and radiomics parameters (CP, PET, CT, PET-CP, CT-CP, and PET-CT-CP) were trained on a subset of patients (75%) using different radiomics software. At baseline (PET/CT0), SUVs, MTV, and TLG were not able to significantly discriminate between patients with a progression of disease or DCB or survival; differently, several radiomics and delta radiomics parameters predicted the outcome with better performance than clinical and conventional PET parameters (AUC > 0.8), with slightly better performance of the parameters extracted from baseline (PET/CT0) and at month 2 (PET/CT1) PET/CT than the delta radiomics parameters. Overall, PET and CT parameters extracted from PET/CT1 were greater predictors than those at baseline. Furthermore, several multivariate models performed well, especially with the radiomics data extracted from PET/CT0 imaging, for both progression prediction (AUC of 1 and 0.96) and DCB (AUC of 0.85 and 0.83 with the PET-CT-CP model).

Recently, Cui et al. [32] performed a prospective study on 30 stage III NSCLC patients (without brain metastasis according to MRI) who received [18F]FDG PET/CT baseline (13 patients) and preoperative scans (29 patients) three weeks after the completion of neoadjuvant treatment (toripalimab + chemotherapy). Lung lesions were delineated by three different nuclear medicine physicians. A total of 6 conventional PET parameters, 102 radiomics features (using the Python package Pyradiomics), and delta features were included in the analysis. The radiological and metabolic responses, in terms of complete pathological response (CPR), were assessed by iRECIST and iPERCIST, respectively; the major pathological response (MPR) was evaluated in the surgical specimen. Twenty patients achieved MPR, and sixteen of them achieved CPR. For delta PET features, the distribution of five SUV statistics features (SUVmax, SUVpeak, SULmax, SULpeak, and TLG) and one radiomics feature (Delta-original-GLDMDependenceNonUniformity—Delta-GLDM-DN) significantly differed both in CPR and non-CPR, MPR, and non-MPR subgroups. No significant correlation between either the radiological or the pathological response or among PD-L1, driver gene status, and baseline PET features was found. At univariate analysis, five SUV parameters and two radiomics features were significantly associated with pathological response, while at the multivariate analysis, SUVmax, SUVpeak, SULpeak, and End-GLDM-LDHGLE were independently associated with CPR. Moreover, SUVpeak and SULpeak performed better than SUVmax and SULmax for MPR prediction. Despite the absence of external validation and the scarce cohort, this study appears remarkable for the homogeneous nature of the evaluated sample and the comprehensive number of PET and clinical data evaluated. Again, the data from the baseline PET resulted as less informative than those obtained from preoperative PET (post-immunotherapy PET), which might provide additional valuable information on the TME and heterogeneity needed to differentiate residual tumor cells and influential immune cells [33].

### 3.4. The Prediction of Adverse Events Correlated with Immunotherapy by [18F]FDG PET/CT and Radiomics

Mu and colleagues [22] evaluated the role of radiomics analysis in predicting the occurrence of irAEs in 146 patients with histologically confirmed advanced-stage (IIIB and IV) NSCLC who underwent [18F]FDG PET/CT 6 months before treatment initiation. Radiomics features extracted from baseline PET, CT, and PET/CT-fusion images were used to generate a radiomics score (RS) to quantify the patient risk for developing irSAE. Indeed, a nomogram model to predict irSAE was developed. The authors found that the radiomics nomogram, incorporating the RS, type of immune checkpoint blockade, and dosing schedule, was able to predict irSAE with an AUC of 0.92 (CI: 0.86, 0.99) and 0.88 (CI: 0.78, 0.97) in the training test and in prospective validation cohorts, respectively.

In another study [27], the same group applied radiomics analysis on [18F]FDG PET/CT images in 175 patients with NSCLC (stage IIIB and IV) to specifically predict the risk of cachexia, a syndrome that induces progressive functional impairment accounting for 20% of cancer-related deaths [38] and promotes primary resistance to ICI. The authors evaluated the durable clinical benefit (DCB; PFS > 6 months) following ICI considering PFS and OS as main endpoints. As a result, the RS was significantly different between cachexic and non-cachexic patients in the training cohort (*p* < 0.001), which was validated in the test cohort (*p* = 0.003) and external test cohort (*p* = 0.04). The RS predicted the risk of cachexia with AUCs ≥ 0.74 in the training, test, and external test cohorts. PFS and OS were significantly shorter among patients with higher radiomics-based cachexia probability in all three cohorts. Furthermore, the RS identified patients with DCB reaching AUCs ≥ 0.66 in all three cohorts. In addition, the authors observed that body mass index (BMI), Eastern Clinical Oncology Group (ECOG), distant metastasis, and RS were significant and independent predictors of cachexia.

## 4. Discussion

### 4.1. Clinical Assessment

As has emerged from the present systematic review, few data are now available about the utility of radiomics in NSCLC candidates or those currently undergoing immunotherapy. Indeed, only 15 papers were selected, and the content for each of them was highly variable.

Some papers discussed the utility of radiomics or deep learning analysis to predict the expression of PD-L1 or to evaluate the TME, with controversial results. Indeed, only one study performed by using deep learning analysis demonstrated an increase in the AUC for the prediction of PD-L1 expression as compared to the standard current criteria.

Moreover, only two papers aimed to predict the response to immunotherapy by using radiomics. However, it is important to have in mind that the complexity of the available treatment landscape and effects on tumor biology may introduce additional challenges and limitations in data analysis and interpretation, which should be acknowledged and addressed in the study design and reporting.

Finally, some efforts have been made for understanding if radiomics can overcome some criticisms in the evaluation of response to immunotherapy. Again, few data are now available, and in two reports, the authors agreed that rather than baseline PET/CT, radiomics analysis from the first scan after the start of immunotherapy or before surgery can be helpful in predicting the success of the therapy.

### 4.2. Radiomics Evaluation

Previous works have suggested that deep learning may achieve better performance than conventional radiomics at the cost, however, of interpretability issues [39,40]. Notably, most of the studies considered in this review (13 out of 15) were based on conventional radiomics and only two on deep learning. This is partly the consequence of deep learning being relatively newer and more complex than conventional radiomics. In addition, whereas significant work has been done towards the standardization of image biomarkers in conventional radiomics [23,41], we cannot affirm the same for deep learning. We should also consider that conventional radiomics is possibly more affordable than deep learning in terms of computational resources and data availability and that the former can rely on user-friendly standalone packages (e.g., LIFEx) that do not require high-end coding skills.

### 4.3. Limitations

One critical point of all the selected studies is that none of them attempted comparing radiomics vs. deep learning in terms of prediction accuracy, interpretability, and/or robustness. Likewise, no studies addressed the possibility of combining the two approaches. Finally, the lack of data sharing, including images, clinical meta-data, and/or code/algorithms, is a limitation that should be addressed in future research.

Independently of their main methodology, all the included studies in this review achieved a low RQS: indeed, none scored above 50%, and most of them averaged around one-third of the available points. The “top-tier” papers were all published by the same group [23,27,38,42], which developed and applied a method to investigate various outcomes of NSCLC patients treated with immunotherapy. The quality issue of radiomics research is well known and has been reported in past reviews [43,44]. Namely, the scores reflect the way RQS is structured, rewarding mainly the prospective studies, as well as the presence of multiple training and validation cohorts. Moreover, many of the RQS checkpoints are tied to a series of procedures related to textural data harmonization and statistical optimization, which were sparsely implemented by the considered studies. Finally, one key point is the level of automation provided by the presented radiomics technique: to obtain the highest score, the method should be able to provide a prediction for the chosen outcome reliably and without human intervention [45]. The presented radiomics analyses are still far from such a lofty goal, even though some of them could be further developed past their current automation level. In general, the requirements of the RQS are stringent; moreover, the second version of the test bench added further requirements and statistical fine-tuning to the score composition. Indeed, striving for data harmonization, statistical integrity, and external validation could be the only way to progress radiomics. In our opinion, three key points should be addressed when developing radiomics research studies. First, multi-center prospective studies should be preferred since they ensure data homogeneity and foster higher external reproducibility of the studies. In this setting, long-term thinking and the allocation of resources should be favored: large, well-planned prospective studies with multiple external validations could provide the most valuable data. The second version of RQS added the international nature of a multi-center study as a further checkpoint, which is a very important issue since the characteristics of the subjects can vary significantly across populations. The scoring is, in fact, affected by the study design (whether it is single-, multi-, or international multi-center) and whether the training dataset comes from different centers as well as the existence of prospective—ideally multi-center—validation arms. Secondly, the key radiomics characteristics in predicting a specific outcome should be identified and their relationship with their biological counterparts investigated [46]. Radiomics “signatures”, while producing an easy-to-read scoring system, offer no insight into the reasons that cause radiomics to be effective in a particular clinical task. Moreover, their results can be hard to reproduce [47,48]. Identifying the key variables (such as entropy, homogeneity, and second-order parameters) could allow linking these characteristics to specific clinical features (e.g., vascularization and tumor response) and even predict the effectiveness of a target therapy in specific cases [29].

### 4.4. Future Perspectives

In the evaluation of TME, an interesting role for radiomics analysis of [18F]FDG PET images could be the assessment of the so called “T-cell exhaustion” phenomenon, which occurs when antigens persist in tumors, and CD8+ T cells exhibit the progressive loss of effector functions and high expression of inhibitory receptors such as PD-1, TIM-3, and many others [49,50], which dampen effector immunity and cause poor responsiveness to ICI therapy. Exhausted T cells have been reported to exhibit metabolic insufficiency, with suppressed glycolysis and restricted glucose uptake [51,52]. It might be assumed that metabolic characteristics of T-cell exhaustion might be reflected by the radiomic features of [18F]FDG-PET images, as already demonstrated by Zhang et al. [14] in a tumor-bearing mice model of lung cancer.

Furthermore, the opportunity to create models that will couple radiomics to more holistic factors such as demographic information (e.g., gender, age, ethnic origin, and geographical location), personal habits (e.g., smoking and occupational exposure), pre-existent clinical conditions (e.g., diabetes, obesity, and COPD), genetic features (e.g., family history, gene expression, and genetic alterations), and tumor biology (e.g., histopathology, immunohistochemistry analysis, and marker expression) will be useful in reducing the selection and imaging performance bias.

In the future, the identified key radiomics features could be parametrized and shown to the clinician as a functional visual map, which could be overlaid on the standard imaging [1]. Such a tool would be pivotal to ensure the transition of radiomics to the clinical setting, with multidisciplinary tools that could be read by all actors participating in the patients’ treatment process, such as radiologists, surgeons, radiation oncologists, nuclear medicine physicians, and medical physicists. Such a path could replicate the success of functional imaging achieved by PET in the latest decades.

## 5. Conclusions

In conclusion, radiomics is promising for the evaluation of TME and for the prediction of response to immunotherapy, but some limitations should be overcome. First of all, the study design should be made by using a specific methodology or criteria. Second, prospective studies are required in order to overcome heterogeneity. Finally, the inclusion in clinical practice of a simple tool able to adequately analyze images has become mandatory for its larger use. Prospective and well-designed studies, by including a large population, are also mandatory.

## Figures and Tables

**Figure 1 cancers-15-03258-f001:**
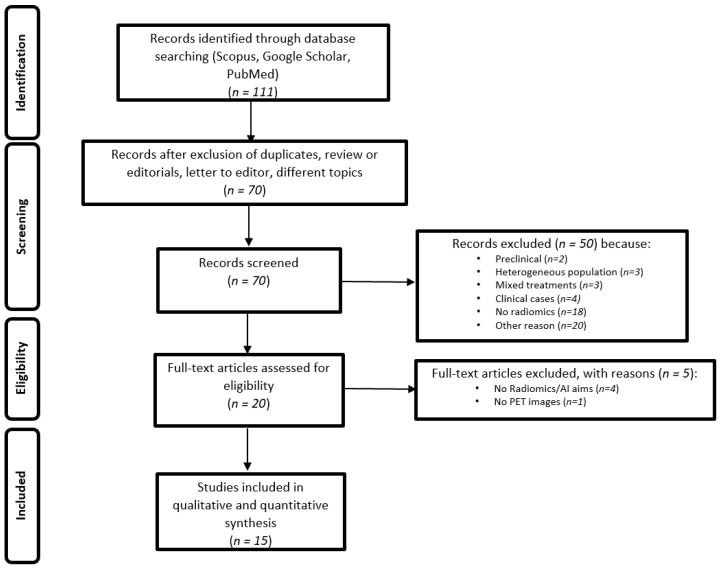
PRISMA statement for the selection of the papers.

**Table 1 cancers-15-03258-t001:** Main characteristics of the selected studies.

Author, Ref	Year of Pub.	Design	Sample Size	Histology	Type of ICIs	Histopathology Correlation	Software	Model	External Validation Cohort	Outcome Measures	Relevant Radiomics Indexes	RQS
Jiang et al. [20]	2019	R	399	NSCLC (SCC and adenocarcinoma)	Atezolizumab and Nivolumab	Yes	ITK V. 3.6.1	Logistic regression and random forest	Na	PD-L1 expression	Shape, IQR, GLCM_ JointAverage, median, NGTDM_contrast	22 (33.3%)
Polverari et al. [21]	2020	R	57	Mixed histologies	Mixed	Yes	LifeX	Univariate analysis	Na	PD-L1 expression; progression status	Coarseness, GLZLM_ ZLNU, kurtosis, skewness, GLZLM_ LZE, GLRLM_RP/SRE/HGRE, GLCM_Homogeneity	13 (19.7%)
Mu et al. [22]	2020	R/P	146 (R), 48 (P)	NSCLC (123 ADC and 71 SCC)	N/S	Yes	In-house software	Logistic regression and Cox multivariate regression	Na	Durable clinical benefit, PFS, and OS	P/R radiomics signatures	28 (42.4%)
Mu et al. [23]	2020	R/P	146 (R), 48 (P)	NSCLC (123 ADC and 71 SCC)	Multiple	Na	In-house software	Multivariable regression analysis	Na	Immune-related adverse events	Radiomic signature (KLD_ SZLGE and KLD_ SRLGE)	26 (39.39%)
Park et al. [24]	2020	R	29	NSCLC (ADC)	Pembrolizumab (10), Nivolumab (18), Atezolizumab (1)	Yes	LifeX v 4	Deep learning	Yes	Cytolytic activity; tumor response, PFS, and OS	N/S	16 (26.23%) *
Valentinuzzi et al. [25]	2020	P	30	NSCLC (17 ADC, 8 SCC, and 5 other)	Pembrolizumab	Na	In-house software	Univariate analysis and Cox regression model	Na	OS	GLRLM_ SRE	22 (33.3%)
Li et al. [26]	2021	R	255	NSCLC (SCC and adenocarcinoma)	N/S	Yes	LifeX v 7	Logistic regression	Na	PD-L1 expression (>1% and >50%)	N/S (12 and 3 feature for >1% and >50%, respectively)	20 (30.3%)
Mu et al. [27]	2021	R	210	NSCLC (109 ADC and 66 SCC)	N/S (anti PD-1 and anti PD-L1)	N	MatLab 2020.a	Uni/multivariable regression analysis	Yes	Cachexia; durable clinical benefit, PFS, and OS	Radiomic signature (SRHGE and LZLGE)	26 (39.39%)
Mu et al. [28]	2021	R/P	648 (R), 49 (P)	NSCLC (531 ADC and 166 SCC)	N/S	Y	ITK	Small residual convolutional network (SResCNN)	Yes	PD-L1 expression; durable clinical benefit, PFS, and OS	N/S	26 (42.6%)
Zhou et al. [29]	2021	R	103	28 SCC and 75 other	N/S	Y	LifeX v 5.1	Univariate analysis and logistic regression	Na	PD-L1 and CD8 expression	GLRLM_ LRHGE, GLZLM_ SZE, SUVmax, NGLDM_Contrast	23 (34.85%)
Tankyevych et al. [30]	2022	R	83	Mixed histologies	Mixed	Y	PyRadiomics	Multivariate model	Na	Survival, progression, and durable clinical benefit	Skewness, median, NGTDM_Complexity, GLCM_Autocorrelation and GLCM_imc1	25 (37.9%)
Tong et al. [31]	2022	R	221	NSCLC (N/S)	N/S	Y	ITK V. 3.8	Clinical-radiomics models; machine learning	Na	CD-8 expression	GLCM_ IMC1, GLSZM_ SZLGE, GLTDM_ LGE, histogram energy, GLTDM_Entropy	24 (36.36%)
Cui et al. [32]	2022	P	29	NSCLC (mixed histologies)	Toripalimab	Y	PyRadiomics	Logistic regression	Na	Pathological response of the primary	Delta SUV-indices; EOT SUV indices; EOT MTV/TLG, EOT uniformity, and EOT GLDM_ LDHGLE	21 (31.82%)
Wang et al. [33]	2022	P	30	NSCLC (16 ADC, 12 SCC, and 2 other)	None **	Y	N/S	Univariate analysis	Yes	Heterogeneity and immune infiltrate	Entropy	16 (24.24%)
Zhao et al. [34]	2023	R	334	NSCLC (163 ADC, 59 SCC, and 112 other)	Pembrolizumab	Y	LifeX v 7	Univariate analysis and logistic regression	Na	PD-L1 expression	GLRLM_RP	20 (30.30%)

Na, not available; * deep-learning-specific scoring was used; ** the correlation with the immune infiltrate suggests that dynamic analysis might be used to evaluate treatment with ICI; ADC, adenocarcinoma; CD-8, cluster of differentiation 8; EOT, end of treatment; GLCM, gray-level co-occurrence matrix; GLDM_LDHGLE, gray-level-dependence matrix_large dependence high gray-level emphasis; GLRLM_RP/SRE/HGRE, grey-level run length matrix_run percentage/short-run emphasis/high gray-level run emphasis; GLSZM, gray-level size-zone matrix; GLTDM, gray-level total displacement matrix; GLZLM, gray-level zone-length matrix; IMC1, informational measure of correlation 1; IQR, interquartile range; KLD_SZLGE/SRLGE, Kullback–Leibler divergence_short-zone gray-level emphasis/short-run low gray-level emphasis; LGE, low gray-level emphasis; LRHGE, long-run high gray-level emphasis; LZE, long-zone emphasis; LZLGE, long-zone low gray-level emphasis; MTV, metabolic tumor volume; N, no; NGTDM, neighborhood grey tone difference matrix; N/S, not specified; NSCLC, non-small cell lung carcinoma; OS, overall survival; P, prospective; PD-L1, programmed death-ligand 1; PFS, progression-free survival; R, retrospective; RQS, radiomic quality score; SCC, squamous cell carcinoma; SRHGE, short-run high gray-level emphasis; SUVmax, maximum standardized uptake value; SZE, short-zone emphasis; SZLGE, short-zone low gray-level emphasis; TLG, total lesion glycolysis; Y, yes; ZLNU, zone-length nonuniformity.

**Table 2 cancers-15-03258-t002:** Radiomics quality score (v 2.0) of the included studies.

Authors (PMID)	Rater			
	F.B.	F.F.	L.M.	Consensus
Jiang et al. [20]	22	22	22	22
Polverari et al. [21]	13	13	15	13
Mu et al. [30]	23	26	26	28
Mu et al. [23]	24	25	23	26
Park et al. [24] *	14	16	15	16
Valentinuzzi et al. [25]	26	27	27	22
Li et al. [26]	20	20	20	20
Mu et al. [27]	27	25	25	26
Mu et al. [28] *	27	27	26	26
Zhou et al. [29]	20	24	20	23
Tankyevych et al. [30]	24	25	23	25
Tong et al. [31]	33	21	30	24
Cui et al. [32]	21	21	21	21
Wang et al. [33]	23	18	18	16
Zhao et al. [35]	27	22	22	20

* deep learning analysis.

## Data Availability

Not applicable.

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
