# Peer review of "PET Radiomics and Response to Immunotherapy in Lung Cancer: A Systematic Review of the Literature"

_cancers, 2023, doi:10.3390/cancers15123258_

Round 1

Reviewer 1 Report

In recent years, more and more attention has been paid to radiomics for its powerful application prospects. In addition to the diagnostic value, it also has predictive effect on the prognosis and treatment response of lung cancer. Immune checkpoint inhibitors (ICIs) are commonly used in lung cancer. It is of great clinical significance to predict the response of lung cancer to immunosuppressive agents by radiomics. On the one hand, it can help rationally select the right patients, on the other hand, it can reduce unnecessary costs.

This manuscript focuses on a comprehensive overview of the existing literature concerning the applications of positron emission tomography (PET)-radiomics in lung cancer patients candidates or undergoing immunotherapy, with clear structure and comprehensive content. This article may be need minormodification before acceptance for publication. My detailed comments are as follows:

1. Please confirm with the editor again whether the format of the review meets the review requirements of the magazine. If not, please modify as required.

2. Line 100: Change “[14][15][16]” to “ [14, 15, 16]” or “ [14-16]”.

3. Table 1: Whether Na is incorrect? please check.

4. Table 2: The two Deep Learning radiomics papers were [21] and [22] respectively, but [41] was makerd with *, it is contradictory.

5. Line 183: (12)?

6. Line 203: Change  [26],[27],[28],[21] to “ [21, 26-28]”

7. Line 216 and 217: Please write numbers in a uniform format.

Minor editing of English language required.

Author Response

In recent years, more and more attention has been paid to radiomics for its powerful application prospects. In addition to the diagnostic value, it also has predictive effect on the prognosis and treatment response of lung cancer. Immune checkpoint inhibitors (ICIs) are commonly used in lung cancer. It is of great clinical significance to predict the response of lung cancer to immunosuppressive agents by radiomics. On the one hand, it can help rationally select the right patients, on the other hand, it can reduce unnecessary costs.

This manuscript focuses on a comprehensive overview of the existing literature concerning the applications of positron emission tomography (PET)-radiomics in lung cancer patients candidates or undergoing immunotherapy, with clear structure and comprehensive content. This article may be need minor modification before acceptance for publication. My detailed comments are as follows:

Q1. Please confirm with the editor again whether the format of the review meets the review requirements of the magazine. If not, please modify as required.

R1. We confirm that the review meets the requirements of the Cancers journal.

Q2. Line 100: Change “[14][15][16]” to “ [14, 15, 16]” or “ [14-16]”.

R2. Done

Q3. Table 1: Whether “Na” is incorrect? please check.

R3. Abbreviation ‘Na’ stands for ‘not available’ in Tab. 1. The correct definition is included in the Table’s legend.

Q4. Table 2: The two Deep Learning radiomics papers were [21] and [22] respectively, but [41] was makerd with “*”, it is contradictory.

R4. We corrected Table 2 accordingly

Q5. Line 183: (12)?

R5. Thanks, the value was deleted.

Q6. Line 203: Change “ [26],[27],[28],[21]” to “ [21, 26-28]”

R6. Done.

Q7. Line 216 and 217: Please write numbers in a uniform format.

R7. Done.

Reviewer 2 Report

Manuscript ID: Cancers-241-5698
Title:  
PET RADIOMICS AND RESPONSE TO IMMUNOTHERAPY  IN LUNG CANCER: A SYSTEMATIC REVIEW OF THE LITERATURE

Date: 2023/06/1

Reviewer's report:
This is an interesting manuscript as it’s a 
comprehensive overview of the existing literature concerning the applications of positron emission tomography (PET)-radiomics in lung cancer patients candidates or undergoing immunotherapy. In recent years, radiomics, an emerging discipline based on the quantitative analysis of imaging data, has been gaining ever-increasing attention for its capability through machine-learning analysis to generate predictive models This MS was one of the few study which systematically review  the possible role of petscan in the evaluation of lung ca patient with the prospect of receiving immunotherapy. I'm sure the result of this study could help in the  decision-making process and guide towards an optimal management of lung ca.

The MS is well prepared and containing a large amount of data.  Although, there remain some limitation.  Nevertheless, it was still well written. However, a few issue  should be clarify prior publication

1.How does this  study determine the predictive and descriptive accuracy as well as the relevancy of the data ?

Author Response

This is an interesting manuscript as it’s a comprehensive overview of the existing literature concerning the applications of positron emission tomography (PET)-radiomics in lung cancer patients candidates or undergoing immunotherapy. In recent years, radiomics, an emerging discipline based on the quantitative analysis of imaging data, has been gaining ever-increasing attention for its capability through machine-learning analysis to generate predictive models. This MS was one of the few study which systematically review  the possible role of petscan in the evaluation of lung ca patient with the prospect of receiving immunotherapy. I'm sure the result of this study could help in the  decision-making process and guide towards an optimal management of lung ca.

The MS is well prepared and containing a large amount of data.  Although, there remain some limitation.  Nevertheless, it was still well written. However, a few issue  should be clarify prior publication

Q1. How does this  study determine the predictive and descriptive accuracy as well as the relevancy of the data?

R1. Thank you for this point. We have implemented and integratedimproved the manuscript with additional considerations to address your question.

Reviewer 3 Report

This review provides a comprehensive overview of the existing literature concerning the applications of positron emission tomography (PET)-radiomics in lung cancer patients candidates or undergoing immunotherapy. Radiomics is a comprehensive analysis methodology for describing tumor phenotypes or molecular biological expressions using minable feature extracted from a large number of medical images. In PET imaging, promising results concerning the ability of handcrafted features to predict the biological characteristics of lesions and to assess patient prognosis or response to treatment have been reported by several authors. In this systematic review, the authors described a current status of PET radiomics in lung cancer. I think this manuscript is interesting, and discussed a hot topic in lung cancer treatment.

In the section of the tumor microenvironment (line238-265), the authors described the PET radiomics and CD8+ tumor infiltrating lymphocytes (TILs). PET radiomics approach is also established to differentiate T cell exhaustion status, which fitted well in immunotherapy. A non-invasive imaging predictor which accurately assessed heterogeneous T cell exhaustion status relevant to immune-checkpoint inhibitor (ICI) treatment, which might estimate immune-responsiveness of tumor microenvironment. Please discuss this issues.

In the discussion section, the text is too long to read. Please divide the text into some subsections, limitation, future direction, etc.

How can we use PET radiomics for a therapy guidance in advanced lung cancer patients in future? We need to avoid selection and performance biases as well as bias through differences in background metabolic activity.

Previous studies with comparable populations included patients with several pretreatments before immunotherapy. How can we evaluate the PET radiomics data of multiple regimens such as immunotherapy combined with chemotherapy as well as those after first line treatment?

In the text, some “PD-L1” was described as “PDL-1”.

There are some typos.

Author Response

This review provides a comprehensive overview of the existing literature concerning the applications of positron emission tomography (PET)-radiomics in lung cancer patients candidates or undergoing immunotherapy. Radiomics is a comprehensive analysis methodology for describing tumor phenotypes or molecular biological expressions using minable feature extracted from a large number of medical images. In PET imaging, promising results concerning the ability of handcrafted features to predict the biological characteristics of lesions and to assess patient prognosis or response to treatment have been reported by several authors. In this systematic review, the authors described a current status of PET radiomics in lung cancer. I think this manuscript is interesting, and discussed a hot topic in lung cancer treatment.

Q1. In the section of the tumor microenvironment (line 238-265), the authors described the PET radiomics and CD8+ tumor infiltrating lymphocytes (TILs). PET radiomics approach is also established to differentiate T cell exhaustion status, which fitted well in immunotherapy. A non-invasive imaging predictor which accurately assessed heterogeneous T cell exhaustion status relevant to immune-checkpoint inhibitor (ICI) treatment, which might estimate immune-responsiveness of tumor microenvironment. Please discuss this issues.

R1. We thank the reviewer for highlighting this interesting topic. Accordingly, we added in the text the following sentences:

In the evaluation of TME, an interesting role for radiomics analysis of [18F]FDG PET images could be the assessment of the so-called “T cell exhaustion” phenomenon, which occurs when antigens persist in tumors and CD8+ T cells exhibit progressive loss of effector functions and high expression of inhibitory receptors, such as PD-1, TIM-3 and so on [27192569, 17950003], which dampen effector immunity and cause poor responsiveness to ICI therapy. Exhausted T cells have been reported to exhibit metabolic insufficiency with suppressed glycolysis and restricted glucose uptake [25809635, 27496729]. It might be assumed that metabolic characteristics of T cell exhaustion might be reflected by radiomics features of [18F]FDG-PET images, as already demonstrated by Zhang et al [36694213] in a tumor-bearing mice model of lung cancer.

Q2. In the discussion section, the text is too long to read. Please divide the text into some subsections, limitation, future direction, etc.

R2. Done

Q3. How can we use PET radiomics for a therapy guidance in advanced lung cancer patients in future? We need to avoid selection and performance biases as well as bias through differences in background metabolic activity.

R3. We thank the reviewer for suggesting this interesting point. Radiomics or artificial intelligence are tools that can help interpret, more in detail, the characteristics of metabolic activity of lung neoplasms at [18F]FDG PET imaging. Indeed, by using specific models, clinical plus imaging plus radiomics, both the likelihood of disease and the evaluation of response to therapy in patients with lung cancer would be better assessed. In the future, we will have the opportunity to create models which will couple radiomics to more holistic factors such as demographic information (e.g., gender, age, ethnic origin, geographical location), personal habits (e.g., smoking, occupational exposure), pre-existent clinical conditions (e.g., diabetes, obesity, COPD), genetic features (e.g., family history, gene expression, genetic alterations) and tumor biology (e.g., histopathology, immunohistochemistry analysis, marker expression), thus reducing the selection and performances biases. We have included a small sentence in the future perspectives accordingly.

Q4. Previous studies with comparable populations included patients with several pretreatments before immunotherapy. How can we evaluate the PET radiomics data of multiple regimens such as immunotherapy combined with chemotherapy as well as those after first line treatment?

R4. Based on the available literature data, only three reports considered the utility of radiomics for evaluating response to immunotherapy. The study by Cui et al, considered adjuvant immunotherapy alone or other adjuvant regimens without reporting the number of patients for each group. Valentinuzzi et al reported the administration of immunotherapy as first-line in 15 patients, second-line in 13 patients and third-line in 2 patients. In only six out of 30 patients, concomitant radiotherapy was used as palliative care. Finally, in the study by Tankyevych et al, immunotherapy was used as a first-line agent in 23 patients and in combination with chemotherapy or targeted therapy in 21 out of 83 patients. Therefore, the high heterogeneity of the data and also the limited number of patients make it difficult to provide a definitive conclusion on how to evaluate the PET radiomics data in case of different therapeutic approaches. However, it is important to consider that the complexity of the treatment landscape and effects on tumor biology that reproduce the real-life scenario may introduce additional challenges and limitations in data analysis and interpretation, which should be acknowledged and addressed in the study design and reporting. A small sentence has been added in the discussion section accordingly.

Q5. In the text, some “PD-L1” was described as “PDL-1”.

R5. Done.

Reviewer 4 Report

This article  conducted a systematic review of lung cancer research  in accordance with the PRISMA guidelines. The language used in the article is also clear and precise, which facilitates understanding and interpretation of the results. And in the introduction, the authors mention the benefits of ICIs treatment in SCLC. However, in subsequent searches, the authors did not specifically target SCLC. For example, in section 2.1, only NSCLC was specifically searched. There is a concern that SCLC patients may have been overlooked, and it is recommended to use keywords that include SCLC when searching for literature to ensure that all relevant studies are included.

The language used in the article is not always concise and could benefit from some editing to improve clarity. For example, “through a mechanism not yet fully known," consider using "via an unknown mechanism."

Author Response

This article  conducted a systematic review of lung cancer research  in accordance with the PRISMA guidelines. The language used in the article is also clear and precise, which facilitates understanding and interpretation of the results. And in the introduction, the authors mention the benefits of ICIs treatment in SCLC. However, in subsequent searches, the authors did not specifically target SCLC. For example, in section 2.1, only NSCLC was specifically searched.

Q1. There is a concern that SCLC patients may have been overlooked, and it is recommended to use keywords that include SCLC when searching for literature to ensure that all relevant studies are included.

R1. We thank the reviewer for spotting this. The study is actually focused on NSCLC, as indicated in the registered protocol (PROSPERO/CRD42023402302). We have therefore removed any reference to SCLC in the revised version of the manuscript.

Q2. The language used in the article is not always concise and could benefit from some editing to improve clarity. For example, “through a mechanism not yet fully known," consider using "via an unknown mechanism."

R2. Done.

Round 2

Reviewer 2 Report

I have seen the revise part, thus, recommend this MS for publication